# Brand Personality of Korean Dance and Sustainable Behavioral Intention of Global Consumers in Four Countries: Focusing on the Technological Acceptance Model

Seung-hye Jung [1,†], Joon-ho Kim [2,†] , Ha-na Cho [3], Hae-won Lee [4] and Hyun-ju Choi [5,*]

1    School of Dance, Kyung Hee University, Seoul 02447, Korea; goldencats_shj@naver.com
2    The Cultural Policy Laboratory, Sangmyung University, Seoul 31006, Korea; kshy4u@naver.com
3    Department of Dance Arts, Hanyang University, Seoul 15588, Korea; artbears@naver.com
4    Department of Dance, Jeonbuk National University College of Arts, Jeonju 54896, Korea;
     moo-2004@hanmail.net
5    Department of Cultural & Arts Management, Sangmyung University, Seoul 31066, Korea
*    Correspondence: hyunju_choi@naver.com
†    First and Second author contributed equally to this paper.

**Abstract:** Brand personality is a useful tool that forms a favorable brand image and that ultimately builds powerful brand equity. However, there has been insufficient empirical research on the brand personality of Korean dance. In the context of using culture and the arts to support national competitiveness, we examine traditional Korean dance in terms of a potential brand personality that can influence the perceptions of global consumers. We look at how this brand can affect consumer perceptions of how easy it is to learn Korean dances as well as their perceptions of the physical benefits of these dances. The respondents included global consumers who had listened to or watched Korean dance music and videos on TV and the Internet, searched for and watched Korean dance videos on YouTube, and searched for Korean dance information on social media at least once. A survey was conducted over the course of four months, from October 2020 to January 2021, in four countries: South Korea, the USA, the UK, and South Africa. Valid data were obtained from 649 individuals. We conducted an empirical study by applying and integrating the technology acceptance model (TAM) to the brand personality of Korean dance. A structural equation model was used to analyze the responses. The brand personality of Korean dance enhanced its perceived ease of use and its perceived usefulness among global consumers, which led to positive attitudes toward the dances. Furthermore, it led to a sustainable behavioral intention, that is, interest in learning traditional Korean dances. Since no studies have integrated Korean dance into a single brand personality to use it as a cultural asset, this study makes considerable contributions to the literature.

**Keywords:** Korean dance; brand personality; technological acceptance model; sustainable behavioral intention; global consumers

## 1. Introduction

Korea's traditional dances represent an important aspect of the country's culture that can be used to spread its influence globally, enhance its national image, and disseminate Korean values [1,2]. Since the beginning of human history, traditional dance has been used as a way of communicating between races that speak different languages [3] and has been the basis of religious ceremonies [4]. Such dances reflect the art and language of personal expression through the body, and, as an aspect of culture, they are the oldest art genre in existence that can be used as a method of social interaction and nonverbal communication [5].

In general, culture has a diverging power [6–8]. Cultural divergence refers to the tendency of a country's culture to flow naturally beyond its borders, similar to water water,

despite heterogeneous environments and can lead citizens in other countries to recognize and appreciate that culture. Diverging cultures not only often influence universal values, such as freedom, equality, and human rights, but also enhance national images and create cultural loyalty beyond national borders [6–8].

In South Korea, the band BTS, who is spearheading K-pop culture, has created a sensation around the world by establishing its own brand personality through its pop songs and Korean-style traditional dances that express Korean culture [9,10]. In the cultural domain, values and brand images are key, spotlighting the importance of symbolic images. From this, we can apply the concept of brand personality.

Currently, Korean dances are considered a traditional cultural asset of South Korea. However, to further their appeal globally, a strategic approach is needed, such as considering Korean dance as a unique artform with its own brand and brand personality. However, although Korean dance has a long history, there have been few studies that have acknowledged its value as a cultural asset or that have investigated its utilization. As there is no single integrated brand personality attached to Korean dance, such a personality is needed to help market it as an important cultural asset of South Korea. Through an integrated brand personality, Korean dance can contribute to the dissemination of Korean culture and the enhancement of its national image [9–11].

Our study attempts the first step of interpreting the brand personality of Korean dance. As Korean dance reflects the beauty and uniqueness of Korean traditional culture, its brand personality can be based on this, which we can then test empirically. Since Korean dance is also an exercise, it has beneficial health effects [12,13]. It is an exercise in which people express emotions with their bodies [14]. Furthermore, Korean dance has a specific charm and an effect that is different from other dance genres because its movements are focused on breathing and not just on music [15]. In general, people believe that Korean dance is "good for health," as its movements help strengthen the core muscles [16]. Moreover, many people can perform its graceful movements, as they are not strenuous or difficult. Korean dance has the potential to be a unique hobby and skill that anyone can enjoy today [17]. In addition to its beneficial exercise effects, it is a unique hobby that is different from other hobbies, as it is associated with various experiences, such as overseas travel [12,13,17].

Currently, foreign tourists (i.e., global consumers) visiting South Korea often try wearing hanbok (i.e., traditional Korean clothes) to experience Korean culture. Koreans also spread their culture when they travel to other countries [18]. In this context, Korean dance can be an embodiment of the national image and a medium to convey Korean culture [9–11].

Based on the above, we assume the following: Korean dance offers ease of use (i.e., the degree of ease of learning it) and perceived usefulness (i.e., the degree of beneficial effects such as diet, posture, walking, muscle strengthening, flexibility, stress relief [19]) to global consumers who have listened to/or watched Korean dance music or videos and who have searched for Korean dance videos on YouTube or for Korean dance information on social media at least once. From this perspective, global consumers may form a positive or negative attitude toward Korean dance, which can lead to behavioral intentions (i.e., wanting to learn Korean dance or recommending it to others [20,21]). Thus, the following linkage is hypothesized: brand personality (Korean dance) → perceived ease of use → perceived usefulness → attitude → behavioral intention (Korean dance). This linkage replicates the theoretical framework of the technology acceptance model (TAM).

The technology acceptance model (TAM) is widely used in studies of the acceptance of information technology (IT) [22–26]. Many previous studies applying the TAM have generally added a large number of independent variables to increase the explanatory power of the research model (e.g., [27–30]). In particular, many studies using the TAM have assessed IT acceptance (e.g., [28,30–32]). These studies adopt simple and convenient research methods and are thus suitable for studying IT.

However, to understand different technological characteristics, it is necessary to add independent variables that can explain those characteristics. Furthermore, studies applying



the TAM are scarce in general social sciences, especially in arts and cultural management. Therefore, to overcome the limitations of previous studies, this study applied the generally used theoretical framework of the TAM to conduct empirical research in arts and cultural management. To this end, as previously mentioned, it set the brand personality of Korean dance as the independent variable and provided the general TAM of brand personality (Korean dance) → perceived ease of use → perceived usefulness → attitude → behavioral intention (Korean dance) as the research model. In sum, this study applied a generally used theory and model to design a TAM for arts and cultural management instead of adding more independent variables.

In sum, despite the long history and tradition of Korean dance, its value and utilization have been low due to a lack of integrated brand personality. Since Korean dance has an integrated identity in terms of the beauty and originality of traditional Korean culture, turning this into a brand personality and empirically testing it is a significant research approach. Accordingly, this study set the following research questions:

RQ1. What is the effect of the brand personality of Korean dance on its perceived ease of use and its perceived usefulness to global consumers?

RQ2. What is the effect of the perceived ease of use on its perceived usefulness to global consumers?

RQ3. What are the effects of the perceived ease of use and the perceived usefulness on the attitudes of global consumers?

RQ4. What is the effect of the attitudes of global consumers on the sustainable behavioral intention to learn Korean dance among global consumers?

The paper has the following six sections: Section 1 (Introduction) describes the research background, motivation, and purpose of the present study. Section 2 (Theoretical Background) examines the theories and presents a literature review of applying brand personality to Korean dance. Section 3 (Method) presents the research model, hypotheses, and design (measurement of the variables, and survey items). Section 4 (Results) examines the characteristics of the research subjects and presents the results of the data analysis, including reliability and validity testing, correlation analysis, and hypothesis testing. Sections 5 and 6 (Discussion and Conclusion) summarize the study and provide implications, limitations, and future research directions.

## 2. Theoretical Background

### 2.1. Brand Personality and Korean Dance

A brand is developed by giving it a personality or human-like characteristics, creating the illusion that the brand is a living object. A strong brand has a personality [33]. Specifically, a brand's personality may be created by using images or representations that impersonate or describe a particular personality. As a concept, it influences consumer attitudes and relationships [34–36]. As a total brand image, it encompasses all of the emotions that consumers have for the brand, including their associations with images, sounds, colors, and smells. People may perceive a brand's personality through direct experiences or indirectly through contact with other people who have used the brand or through advertisements and publicity [34–36].

Brand personality is an important means of differentiating a company's products. In the minds of consumers, a strategically, well-managed brand personality increases brand preference and loyalty. In markets where product performance and quality are standardized, brand personality is increasingly important in terms of competitive positioning [37–39].

Although many studies over the past few decades have investigated brand personality, research on BPS did not begin until the 1990s. Aaker [40] conducted a study of 37 different brands and posited five main dimensions (factors) of brand personality: sincerity, excitement, competence, sophistication, and ruggedness [40–43].

Our study attempts to link a brand personality with Korean dance. Culture and art represent high national brand assets (e.g., awareness, images [44–46]). In an increasingly

advanced society, a country's unique culture and art play a role by enhancing the country's interests and cultural prosperity [47–50]. Therefore, as a cultural asset of South Korea, Korean dance needs to establish a brand personality.

In general, the field of dance allows for longer continuity as a comprehensive artform in terms of communication than other genres and can be considered relatively more useful for indirectly penetrating into and marketing in international markets [51–54]. Therefore, performing arts have important value. Moreover, dance as an art is crucial as a direct or indirect means of cultural dissemination, considering its effects on national image, public relations, and the economy through continuous and gradual strategic support policies [9–11,55,56].

As previously mentioned, a brand's personality is created through symbolism associated with specific values [57,58]. Through this, consumers perceive the brand as a living object based on their experience, which can invoke favorable consumer feelings about the brand [57,58]. In this study, we measure the brand personality of Korean dance by the degrees of sincerity, excitement, competence, sophistication, and ruggedness that are associated with it by global consumers.

### 2.2. Technology Acceptance Model

The TAM was developed as a theoretical framework to identify factors that affect the acceptance of information technology (IT). It was developed as a method to "improve the job performance of the organization" [59–61]. In this context, the TAM is used to focus on the causal relationships among the beliefs, positive/negative attitudes, intention to use, and actual use of a certain innovation of organizational members. Additionally, it can pinpoint external factors that affect the acceptance process [59–61].

The TAM views attitude as a key determinant in predicting a person's intention to use the technology. Furthermore, in the theory of reasoned action, scholars have identified "perceived usefulness" and "perceived ease of use" as determinants of attitudes [62–64]. The TAM defines the perceived usefulness of the technology as "the degree to which a person believes that using a particular system would enhance his/her job performance." Perceived ease of use is defined as "the degree to which a person believes that using a particular system would be free from effort" [62–64]. The TAM also uses the two variables, "perceived ease of use" and "perceived usefulness," as factors that affect user acceptance of new IT innovations; explaining that attitude as a mediator influences behavioral intentions [65–67].

In this study, we define "perceived ease of use" as the perception of the ease of learning Korean dance by global consumers (i.e., the degree to which global consumers believe that learning Korean dance does not require much effort). "Perceived usefulness" is defined as the perception of the beneficial effect (such as diet, posture, walking, muscle strength, flexibility, and stress relief) of Korean dance by global consumers (i.e., the degree to which global consumers believe that performing Korean dance improves health).

### 2.3. Attitude

Currently, the concept of attitude is often defined as "a learned predisposition to respond in a consistently favorable or unfavorable manner with respect to a given object". Attitude is also linked with the concept of perception [68,69]. One can view attitude as a personal preference or non-preference, personal assessment, and behavioral tendency toward a certain idea or object [70]. Among consumers, attitude is a positive or negative assessment of the object. Since attitude has a significant impact on consumer purchasing habits and behavioral intentions, attitude is a crucial factor when studying consumer behavior [71–74]. Moreover, attitude has a strong influence on one's final decision in selecting certain products, services, and brands [75–77]. Attitude can be viewed as an affective component that reflects the overall feeling of a consumer toward a certain object. An affective component combines one's emotions that arise from exposure to a particular stimulus and from exposure to the broader environment [78–80]. In this study, we define attitude as the global consumer's mindset toward Korean dance.

*2.4. Sustainable Behavioral Intention*

Consumers form behavioral intentions primarily based on their perceived emotions toward a product or service and are often based on their experience before and after consumption. This can be described as subjective personal beliefs, expressed in either negative or positive future behaviors, such as revisits to a destination or negative word of mouth about a product [81–83]. In other words, behavioral intentions can be defined as a consumer's feelings toward a product expressed in a future behavior after forming an attitude toward that product [81–83]. That is, a behavioral intention is one's subjective willingness to plan and modify one's future behavior based on the emotion evoked by the product or service or the internal response and experience the person has after the consumption behavior [81,84,85]. Specifically, a behavioral intention is an outcome variable, where making recommendations, for example, or revisiting a website or offering favorable word of mouth are positive sub-variables [86–91].

When the object is a product, positive behavioral intention is expressed as purchase intention, repurchase intention, or reuse intention in the case of the service industry, and revisit intention in the case of tourist destinations [83,88,90,92–95]. The term behavioral intention is comprehensively used in the consumer behavior and service marketing fields [83,96–98].

In this study, behavioral intention includes sustainability, which is considered a major factor in various studies because it directly relates to purchase behaviors (positive word of mouth, revisits, and recommendations [99–101]). Many sustainability studies have examined and discussed consumer responses (e.g., company reviews, purchase intention, and support intention). In certain sectors, "highly sustainable" consumers represent high value-oriented consumption, and, as previously mentioned, strong loyalty (e.g., positive reviews and support intention [102–110]).

In addition, previous studies have examined the demographic characteristics and motivations related to sustainable consumer behavioral intention, examining relevant variables such as decision making, lifestyles, self-identity, and happiness [111–113]. In some cases, the sustainable behavioral intention of consumers manifests in personal values and lifestyles [111,113]. These intentions may appear in several stages of the consumption decision-making process and may be accompanied by various contextual limiting factors or promotional factors [112]. In this study, we define sustainable behavioral intention as the willingness of global consumers to learn Korean dance or recommend it to others.

## 3. Method

*3.1. Research Model*

Based on the research objective, we surveyed global consumers who listened to or watched Korean dance music and videos on TV and the Internet, searched for and watched Korean dance videos on YouTube, and searched for Korean dance information on social media at least once. We then used the data gathered to empirically analyze how the brand personality of Korean dance, as perceived by these global consumers, affected their perceptions of its ease of use and usefulness. We also tested how the perceived ease of use and perceived usefulness affected their attitudes, and how their attitudes affected their sustainable behavioral intentions to learn and/or recommend Korean dance. Our schematized research model is shown in Figure 1.

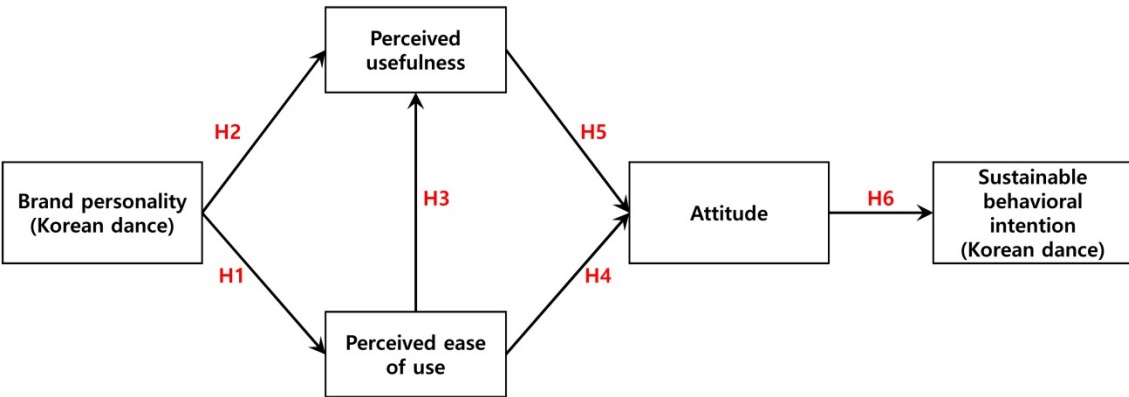

**Figure 1.** Research model.

*3.2. Relationship between Key Variables*

3.2.1. Brand Personality and Perceived Ease of Use/Usefulness

As previously discussed, consumers embrace brand personalities, and a brand's personality strengthens its relationship with consumers [33]. Consumers choose brands related to their own personality and their ideal self and use the brand as a medium for defining their sense of self [34–36].

Associating one's self with a brand is an expression of the consumer's self that they want to communicate with others. In other words, consumers emotionally connect with and convey a brand's personality when they consume a brand [34–36]. Brand personality plays a role in creating a favorable image that consumers want to identify with. A unique brand personality frees the brand from imitations. In the long run, it plays a role in increasing the value of the brand [37–39].

Traditional Korean dance can be seen as having a unique "personality" since it triggers favorable emotions in consumers and creates positive perceptions [57,58]. In fact, Korean dance is unique and is difficult to imitate. As mentioned, beneficial health effects are produced [12,13] as one learns how to express their emotions through dance [14]. It has charm and differs from other genres, as the genre focuses on breathing rather than just on the music [15]. The perception is that it is "good for health," as it strengthens the core muscles [16] through easy movements. It has the potential to be a unique hobby and skill that people can enjoy today [17].

In this study, to apply the TAM, the ease of learning Korean dance as perceived by global consumers, corresponds to "perceived ease of use", and its beneficial effects (e.g., diet, posture, walking, muscle strength, flexibility, stress relief) as perceived by global consumers correspond to "perceived usefulness." Our assumption is that the perceived brand personality of Korean dance will improve the favorable emotions and perceptions of global consumers, which, in turn, will improve its perceived ease of use and perceived usefulness. Therefore, we posit the following hypotheses:

**Hypothesis 1 (H1):** *The brand personality of Korean dance will have a positive effect on its perceived ease of use.*

**Hypothesis 2 (H2)** : *The brand personality of Korean dance will have a positive effect on its perceived usefulness.*

3.2.2. Perceived Ease of Use/Usefulness and Attitude

Many previous studies have verified the influential relationships between perceived ease of use, perceived usefulness, and attitude, key elements of the TAM. Recently, for example, Indarsin and Ali [114] conducted a study on attitudes toward m-commerce and showed that perceived usefulness had a strong impact on the attitude toward m-commerce, while perceived ease of use had a moderate impact on attitudes toward m-commerce.

Hansen et al. [115] showed that perceived ease of use had a statistically significant positive effect on perceived usefulness in terms of predicting consumer social media use for transactions.

Ma et al. [62] conducted a perception study of perceived ease of use and the perceived usefulness of sustainability labels on apparel products. Their analysis showed that perceived ease of use and perceived usefulness had significant impacts on attitude and purchase intentions, confirming that they related to consumer use of sustainability labels. Raza et al. [116] investigated new determinants of perceived ease of use and perceived usefulness for mobile banking adoption and showed that perceived ease of use had a statistically significant impact on perceived usefulness. Meanwhile, perceived ease of use and perceived usefulness had significantly positive causal relationships with attitude. In brief, they found that perceived ease of use, perceived usefulness, and user attitudes had positive effects on mobile banking adoption.

In a TAM-based study on the use intention of multimedia teaching materials among schoolteachers, Weng et al. [67] showed that the perceived ease of use related to the IT environment had a positive effect on their perceived usefulness. This, in turn, had a significant impact on user attitudes toward the teaching materials. Finally, positive user attitudes increased their intention to use these teaching materials. With respect to the e-purchasing intentions of consumers, Moslehpour et al. [117] used personality characteristics (i.e., conscientiousness and openness) as independent variables to examine the causality between perceived ease of use and perceived usefulness. Their analysis confirmed that there were positive correlations in the relationships between consumer e-purchasing intentions, perceived ease of use, and perceived usefulness. Based on these studies, we posit the following hypotheses:

**Hypothesis 3 (H3):** *The perceived ease of use of Korean dance will have a positive effect on its perceived usefulness.*

**Hypothesis 4 (H4):** *The perceived ease of use of Korean dance will have a positive effect on attitudes toward it.*

**Hypothesis 5 (H5):** *The perceived usefulness of Korean dance will have a positive effect on attitudes toward it.*

3.2.3. Attitude and Behavioral Intention

Many studies have also examined the relationship between attitude and behavioral intention. For example, recently, Yeo et al. [83] investigated the relationships between consumer experiences, attitudes, and behavioral intentions toward online food delivery (OFD) services. They showed that consumer experiences with OFD services had a positive impact on their attitudes, which ultimately increased behavioral intention. Choe and Kim [81] conducted empirical tests on the value of the effects of tourist local food consumption on attitude, food destination image, and behavioral intention. Their analysis showed that tourist attitudes toward local food had a positive effect on the food destination image, and both the attitudes and food destination image had a positive effect on the behavioral intentions of tourists.

In a study on the determinants of e-waste recycling behavioral intentions, Thi Thu Nguyen et al. [118] demonstrated that environmental awareness and the attitudes of local residents toward recycling had direct impacts on behavioral intentions. Yen et al. [119] conducted a study on the predictive factors of behavioral intentions to use urban green spaces. The results revealed that positive attitudes toward the use of urban green spaces increased behavioral intentions (e.g., a good place to socialize/visit/rest, and intentions to visit frequently). Based on the above, we posit the following:

**Hypothesis 6 (H6):** *Attitude will have a positive effect on the sustainable behavioral intention to learn Korean dance.*

### 3.3. Variables

Table 1 lists the specific items measured in the survey. There are 23 questions for the survey measurement items of all of the variables, which are measured on a five-point Likert scale (1 = strongly disagree, 5 = strongly agree).

**Table 1.** Survey measurement items of variables.

| Variables | Operational Definition | Measurement Items | Source |
|---|---|---|---|
| Brand Personality of Korean Dance | Consumer perception of its brand identity (e.g., sincerity, excitement, competence, sophistication, and ruggedness). | Korean dance—seems to be sincere (honest) and familiar. | Coelho et al. [57] Pradhan et al. [120] Priporas et al. [121] |
| | | Korean dance—seems to be exciting and unique. | |
| | | Korean dance—seems to be competent and intelligent. | |
| | | Korean dance—seems to be sophisticated and attractive. | |
| | | Korean dance—seems to be rugged and extroverted. | |
| Perceived Ease of Use | Consumer perception of its ease of learning (i.e., Korean dance does not require much effort). | Korean dance—beginners can learn it easily and comfortably. | Chen and Aklikokou [122] Gupta et al. [123] Wicaksono and Maharani [124] |
| | | Korean dance—beginners can learn it without difficulty. | |
| | | Korean dance—beginners can learn it with little effort. | |
| | | Korean dance—beginners can approach and learn it easily. | |
| Perceived Usefulness | Consumer perception of its beneficial effects (e.g., diet, posture, walking, muscle strength, flexibility, stress relief). | Korean dance—is helpful for diet, compared with other dances. | Chen and Aklikokou [122] Gupta et al. [123] Wicaksono and Maharani [124] |
| | | Korean dance—is helpful for posture/walking correction, compared with other dances. | |
| | | Korean dance—is helpful for enhancing muscle strength/flexibility, compared with other dances. | |
| | | Korean dance—is helpful for stress relief, compared with other dances. | |
| Attitude | The positive consumer mindset toward Korean dance. | Korean dance—is more enjoyable than expected. | Chawla and Joshi [125] Meng et al. [126] Zarei et al. [127] |
| | | Korean dance—is more captivating than expected. | |
| | | Korean dance—I am attached to it more than expected. | |
| | | Korean dance—I have more affection for it than expected. | |
| | | Korean dance—I have more positive feelings about it than expected. | |
| Behavioral Intention | The degree to which consumers want to learn and recommend Korean dance. | Korean dance—I will positively consider learning it. | Choi et al. [11] Kwak et al. [9] Meng et al. [126] |
| | | Korean dance—I will recommend it to others. | |
| | | Korean dance—I will talk to others positively about learning it. | |
| | | Korean dance—I will choose to learn it over other dances. | |
| | | Korean dance—I am thinking of learning it in the future. | |

Note: There are a total of 23 questions for the survey measurement items for all variables. All variables were measured on a five-point Likert scale (1 = *strongly disagree* to 5 = *strongly agree*). We used previous studies as references for the survey measurement items for all variables and modified/supplemented them according to our intent. Furthermore, we developed some questions of our own.

## 4. Results

### 4.1. Data Analysis

We used SPSS and SmartPLS for statistical analysis. The data analysis procedure was as follows: We conducted a frequency analysis to examine the demographic characteristics, a reliability analysis using Cronbach's alpha coefficient to test the reliability of the measurement items, and a factor analysis to test validity. Furthermore, we conducted a correlation analysis to examine the degree of closeness (i.e., correlation) between the variables. Finally, we built a structural equation model to test the causality between variables—the essence of the study.

### 4.2. Respondents

The survey was conducted through Netpoint Enterprise Inc. (http://www.netpoint.co.kr/ (accessed on 6 October 2021)), a global research organization, over four months, from 1 October 2020 to 31 January 2021. The questionnaire was created in two languages: Korean and English. From the survey, we collected a total of 649 valid samples from four countries: South Korea, the USA, the UK, and South Africa. Table 2 presents the respondent demographics.

**Table 2.** Demographic characteristics ($N$ = 649).

| Items | | Frequency | % |
|---|---|---|---|
| Gender | Male | 326 | 50.2 |
| | Female | 323 | 49.8 |
| Age | 20s | 103 | 15.9 |
| | 30s | 178 | 27.4 |
| | 40s | 179 | 27.6 |
| | 50s | 189 | 29.1 |
| Education | High school | 106 | 16.3 |
| | Technical college | 120 | 18.5 |
| | University | 328 | 50.5 |
| | Graduate school | 95 | 14.6 |
| Monthly income (personal) | KRW 2 million or less | 207 | 31.9 |
| | KRW 2.01 million–3 million | 133 | 20.5 |
| | KRW 3.01 million–4 million | 100 | 15.4 |
| | KRW 4.01 million–5 million | 81 | 12.5 |
| | KRW 5.01 million or greater | 128 | 19.7 |
| Nationality | South Korea | 208 | 32.0 |
| | USA | 143 | 22.0 |
| | UK | 139 | 21.4 |
| | South Africa | 159 | 24.5 |

### 4.3. Reliability and Validity

We conducted reliability and validity analyses related to the survey measurement items of all of the variables that were used. Table 3 shows the results in detail. The alpha coefficient is at least 0.894 for all variables, indicating high reliability. The loading value of each factor is at least 0.821 as well. Furthermore, the average variance extracted (AVE) value is at least 0.741. Hence, we verified both the reliability and validity of the survey measurement items for all of the variables that were used.

**Table 3.** Reliability and validity.

| Variables | Items | Convergent Validity | | | Cronbach's Alpha | Multi-Collinearity |
| | | Outer Loadings | Composite Reliability | AVE | | VIF |
|---|---|---|---|---|---|---|
| Brand personality | Brand personality 1 | 0.849 | 0.935 | 0.741 | 0.913 | 2.518 |
| | Brand personality 2 | 0.870 | | | | 2.905 |
| | Brand personality 3 | 0.876 | | | | 2.886 |
| | Brand personality 4 | 0.887 | | | | 3.162 |
| | Brand personality 5 | 0.821 | | | | 2.102 |
| Perceived ease of use | Perceived ease of use 1 | 0.904 | 0.956 | 0.845 | 0.939 | 3.450 |
| | Perceived ease of use 2 | 0.938 | | | | 4.783 |
| | Perceived ease of use 3 | 0.937 | | | | 4.668 |
| | Perceived ease of use 4 | 0.898 | | | | 3.052 |
| Perceived usefulness | Perceived usefulness 1 | 0.850 | 0.927 | 0.759 | 0.894 | 2.168 |
| | Perceived usefulness 2 | 0.887 | | | | 2.751 |
| | Perceived usefulness 3 | 0.878 | | | | 2.641 |
| | Perceived usefulness 4 | 0.870 | | | | 2.423 |
| Attitude | Attitude 1 | 0.872 | 0.950 | 0.791 | 0.934 | 2.979 |
| | Attitude 2 | 0.878 | | | | 3.061 |
| | Attitude 3 | 0.893 | | | | 3.336 |
| | Attitude 4 | 0.905 | | | | 4.237 |
| | Attitude 5 | 0.899 | | | | 3.766 |
| Sustainable behavioral intention | Sustainable behavioral intention 1 | 0.893 | 0.952 | 0.800 | 0.937 | 3.285 |
| | Sustainable behavioral intention 2 | 0.908 | | | | 3.806 |
| | Sustainable behavioral intention 3 | 0.883 | | | | 3.107 |
| | Sustainable behavioral intention 4 | 0.895 | | | | 3.741 |
| | Sustainable behavioral intention 5 | 0.892 | | | | 3.660 |

Notes: Measurement item: five-point Likert scale (1 = *strongly disagree* to 5 = *strongly agree*). Outer loadings > 0.70. Composite reliability > 0.70. Average variance extracted (AVE) > 0.5. Cronbach's alpha > 0.70. Variance inflation factor (VIF) < 10.0.

### 4.4. Correlation Analysis

Table 4 shows the results of our discriminant validity analysis (i.e., correlation analysis). We examined whether the square root of the AVE exceeds the correlation coefficient between each variable. The results show that the square root of the AVE is higher than the correlation coefficient between each variable in every case, thus verifying the discriminant validity of the variables that were used.

**Table 4.** Correlation analysis.

| Variable | Brand Personality | Perceived Ease of Use | Perceived Usefulness | Attitude | Sustainable Behavioral Intention |
|---|---|---|---|---|---|
| Brand personality | **0.861** | | | | |
| Perceived ease of use | 0.455 | **0.919** | | | |
| Perceived usefulness | 0.700 | 0.670 | **0.871** | | |
| Attitude | 0.679 | 0.647 | 0.756 | **0.889** | |
| Sustainable behavioral intention | 0.553 | 0.691 | 0.715 | 0.786 | **0.894** |

Note: The diagonal elements in bold are the respective square root of the average variance extracted.

### 4.5. Hypothesis Testing

To test our hypotheses, we used structural equation modeling (SEM) in SmartPLS. Resampling was performed 500 times using the bootstrapping method [9,10,55,128,129]. Bootstrapping is a non-parametric procedure that can test the statistical significance of various partial least squares regression-SEM results, such as the path coefficients, Cronbach's alpha, HTMT (heterotrait-monotrait monotrait), and $R^2$ values [9,10,55,128,129]. Table 5 presents the results of the analysis.

**Table 5.** Hypothesis testing results.

| | Path | | | β-Value | Sample Mean | Standard Deviation | *t*-Value | *p*-Value | Hypothesis |
|---|---|---|---|---|---|---|---|---|---|
| H1 | Brand personality | → | Perceived ease of use | 0.455 | 0.457 | 0.035 | 12.968 | 0.001 | Supported |
| H2 | Brand personality | → | Perceived usefulness | 0.498 | 0.498 | 0.030 | 16.461 | 0.001 | Supported |
| H3 | Perceived ease of use | → | Perceived usefulness | 0.444 | 0.445 | 0.029 | 15.059 | 0.001 | Supported |
| H4 | Perceived ease of use | → | Attitude | 0.255 | 0.256 | 0.038 | 6.649 | 0.001 | Supported |
| H5 | Perceived usefulness | → | Attitude | 0.585 | 0.584 | 0.037 | 15.860 | 0.001 | Supported |
| H6 | Attitude | → | Sustainable behavioral intention | 0.786 | 0.788 | 0.018 | 44.337 | 0.001 | Supported |

The brand personality of Korean dance (β = 0.455, *t* = 12.968, *p* < 0.01) has a statistically significant positive effect on its perceived ease of use and (β = 0.498, *t* = 16.461, *p* < 0.01) on its perceived usefulness. Furthermore, its perceived ease of use (β = 0.444, *t* = 15.059, *p* < 0.01) has a statistically significant positive effect on perceived usefulness. Therefore, Hypotheses 1, 2, and 3 are all supported.

Both perceived ease of use (β = 0.255, *t* = 6.649, *p* < 0.01) and perceived usefulness (β = 0.585, *t* = 15.860, *p* < 0.01) have a statistically significant positive effect on attitude. Therefore, Hypotheses 4 and 5 are supported.

Attitude toward Korean dance (β = 0.786, *t* = 44.337, *p* < 0.01) has a statistically significant positive effect on the sustainable behavioral intention to learn Korean dance. Therefore, Hypothesis 6 is supported.

### 4.6. Comparison of Nationalities of Global Consumers

As mentioned above, this study conducted a hypothesis test for the entire group. However, since there may be statistically significant differences among the nationalities of global consumers, an additional analysis was conducted. As shown in Table 6, the results of the statistical analysis among global consumers in four countries were the same as that for the entire group. That is, it was confirmed that there was a statistically significant effect in all pathways, and no differences occurred.

**Table 6.** Comparison among nationalities.

| | Path | | | South Korea | | USA | | UK | | South Africa | |
|---|---|---|---|---|---|---|---|---|---|---|---|
| | | | | β | *t* | β | *t* | β | *t* | β | *t* |
| H1 | Brand personality | → | Perceived ease of use | 0.342 | 5.323 ** | 0.528 ** | 7.021 ** | 0.552 ** | 7.549 ** | 0.310 ** | 4.067 ** |
| H2 | Brand personality | → | Perceived usefulness | 0.491 | 9.001 ** | 0.445 ** | 6.309 ** | 0.536 ** | 9.539 ** | 0.516 ** | 10.411 ** |
| H3 | Perceived ease of use | → | Perceived usefulness | 0.405 | 8.122 ** | 0.538 ** | 9.416 ** | 0.428 ** | 7.086 ** | 0.414 ** | 6.922 ** |
| H4 | Perceived ease of use | → | Attitude | 0.204 | 3.383 ** | 0.311 ** | 4.336 ** | 0.339 ** | 3.591 ** | 0.256 ** | 3.182 ** |
| H5 | Perceived usefulness | → | Attitude | 0.596 | 9.666 ** | 0.625 | 8.340 ** | 0.480 ** | 5.387 ** | 0.571 ** | 7.169 ** |
| H6 | Attitude | → | Sustainable behavioral intention | 0.695 | 17.629 ** | 0.850 ** | 27.591 ** | 0.828 ** | 25.635 ** | 0.743 ** | 17.910 ** |

Note: ** *p* < 0.01.

## 5. Discussion

### 5.1. Summary of Research

In the study, we empirically analyzed the effects of the Korean dance brand on its perceived ease of use and perceived usefulness among global consumers. Furthermore, we tested the effects of its perceived ease of use and perceived usefulness on the attitudes of global consumers and the effects of these attitudes on sustainable behavioral intentions to learn Korean dance. A structural equation model was used to analyze the responses.

(1) The results show that the brand personality of Korean dance has a statistically significant positive effect on both its perceived ease of use and its perceived usefulness; perceived ease of use also has a statistically significant positive effect on perceived usefulness. Our results underpin those in previous studies by Chang [14], Choi and Jung [17], Choi and Jung [12], Choi et al. [13], Hansen et al. [115], Kim et al. [15], Park et al. [16], Raza et al. [116], and Weng et al. [67].

Consumer involvement with various brands may vary depending on personal relevance, familiarity, and experience with the brand [130,131]. In a situation where various brands compete, consumers may evaluate a certain brand more rationally or emotionally than others [132,133]. If consumers emotionally respond to a brand's personality, the brand may become part of the consumer's self-expression [134–136], as the brand's personality helps consumers define and express their own identity [137,138].

Based on our results, we can say that when watching or learning Korean dances in person, global consumers feel a sense of unity with these dances, as if they reflect their sense of self; thus, the brand personality of Korean dance creates a stable and long-term relationship with the consumer. Global consumers can listen to or watch Korean dance music and videos with just a few clicks on the Internet. Moreover, they can easily search for and watch Korean dance videos on social media and YouTube, among other channels. Through indirect experience with various Korean dances on the Internet, global consumers may develop emotions and feelings corresponding to the brand's personality. It seems that these global consumer emotions have positive effects on the perceived ease of use of the dances (i.e., ease of learning Korean dances) and their perceived usefulness (i.e., beneficial effects of Korean dances).

(2) Both its perceived ease of use and its perceived usefulness had statistically significant positive effects on attitude. These results corroborate those of previous studies by Indarsin and Ali [114], Ma et al. [62], Raza et al. [116], and Weng et al. [67]. Therefore, this study confirmed that global consumers form positive attitudes toward Korean dance (positive mindset toward Korean dance) when they rate its perceived ease of use and perceived usefulness positively.

(3) Attitude had a statistically significant positive effect on sustainable behavioral intention to learn Korean dance. These results align with those of previous studies by Choe and Kim [81], Thi Thu Nguyen et al. [118], Yen et al. [119], and Yeo et al. [83]. Thus, we confirm that when global consumers form a positive attitude toward Korean dance, it ultimately leads to sustainable behavioral intentions to learn the dances (i.e., the continuous desire to learn Korean dance or recommend it to others).

### 5.2. Research Implications

#### 5.2.1. Business Implications

This study empirically proved that global consumers form positive attitudes toward Korean dance, which lead to sustainable behavioral intentions such as learning Korean dance and spreading information about Korean dance through word of mouth. Therefore, this study provides the following business implications:

It is necessary to introduce and expose Korean dance performances to local markets to inform the world about Korean dance. One example would be to provide an occasion to watch Korean dance performances offline in local markets and to promote or advertise the appeal of Korean dance via major local broadcasting media. It is also necessary to maximize the publicity of Korean dance performances by having popular Hallyu stars

make an appearance in each inbound market. However, putting this into practice is currently difficult because of the COVID-19 pandemic. By publicizing or advertising Korean dance in the local media, global consumers would form positive attitudes toward Korean dance, which would lead to sustainable behavioral intentions. Moreover, it is necessary to implement a tour program so that international tourists (global consumers) visiting Korea can enjoy Korean dance performances.

5.2.2. Theoretical/Social Implications

(1) This study is the first to attach a brand personality to Korean dance and to apply the TAM in a cultural and artistic context. Despite its long history and tradition, few studies have looked at Korean dance as an integrated cultural brand. This study is significant in that it identifies a brand personality for Korean dance as perceived by global consumers and empirically tests it by using the beauty and uniqueness of the dance to simulate an integrated brand personality for it.

(2) As previously mentioned, studies applying the TAM in general social sciences are rare, especially in the arts and cultural management. Therefore, this study contributes to the literature by conducting empirical research on arts and cultural management as well as by applying the generally used theoretical framework of the TAM to overcome the limitations of previous studies. In particular, this study set the brand personality of Korean dance as the independent variable and provided the general TAM of perceived ease of use → perceived usefulness → attitude → behavioral intention (Korean dance) as the research model. Thus, researchers can use its findings to design the TAM and apply the theory in the arts and cultural management.

(3) There is still insufficient social awareness on the art of dancing. In particular, there is a lack of awareness of social dance (including dance for all) in Korea due to the dogmatism of elite dance. In general, the predominant view of Korean dance is that it is exclusive to a few special people. However, there should be a change in perception to it as a dance that everyone can easily learn and enjoy. The more people think that Korean dance is easy to learn, the more they will want to learn Korean dance and recommend others to learn it as well. Furthermore, people with more faith in the effect of Korean dance in correcting posture, strengthening muscles, and relieving stress tend to show a better attitude and higher sustainable behavioral intentions. Therefore, through Korean dance, global consumers can not only understand a culture that is not their own, but they can also understand all eras of humanity. This indicates that Korean dance is a good way to understand, respect, and learn about other cultures. In other words, by improving the brand personality of Korean dance, global consumers can better understand Korean dance from a social perspective.

(4) Modern society has improved quality of life thanks to globalization, informatization, and openness, allowing people to enjoy a variety of cultural benefits. Conversely, the introduction of Western culture has gradually dissipated the understanding of traditional Korean culture as well as national identity. However, advanced countries hold a dominant position in cultural competitiveness by preserving and developing their traditional culture and seeking globalization. Therefore, this study implies the need for Korea to pursue the globalization of traditional culture through Korean dance. Here, tradition refers to the independent brand personality of national culture. This is not limited to the cultural values of the past that generations have been passed down over time. In other words, the tradition continues to exist as existential cultural values and even aims for future cultural values. Among them, traditional Korean culture forms the basis of Korean dance with traditional Korean aesthetics. Therefore, Korea must enhance the brand personality of Korean dance—one form of traditional Korean culture—and take the initiative in globalizing Korean dance.

(5) The results enable predictions related to the Korean dance performance industry as well as other cultural and art experience products and services. That is, we established that global consumers from the four countries (South Korea, the USA, the UK, and South Africa)

formed certain preferences through repetitive experiences, such as watching/learning Korean dance performances. Therefore, other potential cultural art service products for global consumers can be developed and measured in the future. Furthermore, the production and consumption of Korean dance can occur simultaneously within a concert hall. By maximizing the brand personality of Korean dance, a multi-use single-source for the brand could be used to disseminate Korean dance content for performance/learning in the future. By taking advantage of the Korean Wave, the Korean dance industry can penetrate more global markets.

*5.3. Research Limitations and Future Directions*

Although this study has several constructive implications, it has the following limitations: (1) The respondents were global consumers who had listened to, watched, or searched for Korean dance music and videos on different media channels at least once. However, we expect that global consumers who have visited South Korea and who have watched Korean dance in a concert hall or who have learned Korean dance in person at least once might respond differently to its perceived brand personality from those who have seen Korean dance only online. Therefore, a follow-up study should include a comparative analysis with other groups. Specifically, a comparative analysis will be needed between groups who have encountered and experienced a Korean dance performance at least once online and groups that have experienced a Korean dance performance offline at least once.

(2) This study conducted a survey on global consumers in only four countries: Korea, the United States, the United Kingdom, and the Republic of South Africa. However, there may be some limitations in generalizing the research results since the statistical analysis was performed on only four countries. Furthermore, this study limited the outcome variables only to attitude and sustainable behavioral intentions. Therefore, it is necessary to derive detailed analysis results by drawing loyalty variables in more detail, such as global consumer satisfaction, trust, word of mouth, and so on.

(3) Based on the structure of the brand personality affecting perceived ease of use, perceived usefulness, attitude, and sustainable behavioral intention, that is, the structure proposed in this study, there may be other potential variables that have an effect on sustainable behavioral intentions that should be added in the future.

## 6. Conclusions

This study adopted the concept of potential brand personality, which may affect global consumer awareness of traditional Korean dance, thereby improving national competitiveness and integrating culture and art. We examined how this can affect the awareness of global consumers about the ease of learning Korean dance and the physical benefits of dancing through brand personality. Since no studies have integrated Korean dance into a single brand personality to use it as a cultural asset, this study makes considerable contributions to the literature. Furthermore, it is the first piece of empirical research that has examined the causal relationships among brand personality, perceived ease of use, perceived usefulness, attitude, and behavioral intention of Korean dance. Therefore, policymakers can use this study's significant and useful contributions as a reference for strategies and policies related to the Korean dance performance industry as well as other products and services in the arts and cultural experiences.

**Author Contributions:** The authors contributed equally to this work. All of the authors contributed to the conceptualization, formal analysis, investigation, methodology, writing of the original draft, and the review and editing. All authors have read and agreed to the published version of the manuscript.

**Funding:** This research received no specific grant from any funding agency in the public, commercial, or not-for-profit sectors.

**Institutional Review Board Statement:** Ethical review and approval were waived for this study because although it was a human study, it was observational, and the research design did not involve ethical issues.

**Informed Consent Statement:** Informed consent was obtained from all subjects involved in the study.

**Data Availability Statement:** Data sharing is not applicable. The data are not publicly available due to participant privacy.

**Conflicts of Interest:** The authors declare no conflict of interest.

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
