# Peer review of "Brand Personality of Korean Dance and Sustainable Behavioral Intention of Global Consumers in Four Countries: Focusing on the Technological Acceptance Model"

_sustainability, doi:10.3390/su132011160_

Round 1

Reviewer 1 Report

The manuscript presents an interesting perspective upon the perceptions of Korean dance and brand personality. Overall, the article is well written and presents a significant amount of data and citations which are relevant to the context of the study. I would have appreciate a clearer outline/summary of the methodology/hypothesis in the introductory paragraphs and I would also recommend that further breakdown of the 'data' is explained/described in support of the information presented in charts/visual format. The introduction positions culture as a core component of the study and I would  have liked to read how the authors contend with this as a result of their investigations more explicitly in the concluding paragraphs.   

Author Response

I completed the response to the “Reviewer Comments”, as you have requested.

The “Reviewer Comments” file is uploaded on the journal website.

Reviewer 2 Report

The article offers an original analysis of traditional Korean dance using the concept of brand personality, seeing it as having a unique “personality”, that triggers favorable emotions in consumers and creates positive perceptions. The study examined in detail and measured empirically the brand personality of Korean dance and tested how perceived ease of use and perceived usefulness affected the attitudes of global consumers and their sustainable behavioral intention to learn and recommend Korean dance.

The article is precisely framed with clearly defined concepts and categories, as well as correctly interpreted and presented results.

I believe the article will be of interest to a wide audience.

I think the manuscript is well structured and well written.

The abstract clearly reflects the objectives of the studies and main findings.

The Introduction also is a very interesting in its orientation to display the traditional Korean dance in the context of the brand personality scale (BPS).

The chosen research methods are appropriate and correspond to the research questions.

The study attempts to link a brand personality with Korean dance and with the concept of 'perceived usefulness' of the Technology Acceptance Model (TAM) to define “perceived ease of use” as the perception of the ease of learning Korean dance.

The literature review and hypothesis development are relevant.

The aim of the study was to test how perceived usefulness can affected the attitudes of global consumers and their sustainable behavioral intention to learn and recommend Korean dance.

The results and conclusion are clearly presented and argumented.

Author Response

(The authors gave the same response as above.)

Reviewer 3 Report

The article it is interesting as a cross cultural model.

But the authors must to make some changes:

Abstract more about the survey structure, and redesign it focusing on culture.

Line 253

3 Literature Review and Hypothesis Development

Here there are a mix of ideas so will be beneficial for readers to let only the literature so the authors must improve that section.

Line 343

4.1. Research Model 343

Will be better if you authors will include here the hypothesis to be more explicit for readers

Also put the entire hypothesis H1….H5 together in Figure 1 are 6 hypotheses H6 please verify.

You forgot to put as study case the target from abstract and the period also. So the authors are not following a structure of article so first they have to present the

  1. Target
  2. Period
  3. Survey
  4. Model of research
  5. Results

In article it is a mix.

Line 363

4.3. Respondents

You have to delete and put the Line 364-371 to the section with model of research or study case.

We surveyed global consumers who listened to or watched Korean dance music 364 and videos on TV and the Internet, searched and watched Korean dance videos on 365 YouTube, and searched for Korean dance information on social media at least once. The 366 survey was conducted through Netpoint Enterprise Inc. (http://www.netpoint.co.kr/), a 367 global research organization, over four months from October 1, 2020 to January 31, 2021. 368 The questionnaire was created in two languages: Korean and English. From the survey, 369 we collected a total of 649 valid samples from four countries: South Korea, the USA, the 370 UK, South Africa.

Line 371 : Table 2 presents respondent demographics. 

Move them in Section RESULTS  here you will put all the results obtain.

Line 374 -381

4.4 Data Analysis

We used SPSS and SmartPLS for statistical analysis. The data analysis procedure 375 was as follows. We conducted a frequency analysis to examine the demographic 376 characteristics, a reliability analysis using Cronbach’s alpha coefficient to test the 377 reliability of the measurement items, and a factor analysis to test validity. Furthermore, 378 we conducted a correlation analysis to examine the degree of closeness (i.e., correlation) 379 between variables. Finally, we built a structural equation model to test the causality 380 between variables—the essence of the study.

Again you have to put it before the result

Conclusion they have to include the dance as a component of culture not a single component

References must to be adapted with new research

In conclusion the article must to be reorganized under scientific rules with a logical idea and presentation.

Author Response

I completed the response to the “Reviewer Comments”, as you have requested.

The “Response to Review Comments” file is uploaded on the journal website.

Reviewer 4 Report

I am pleased to have the opportunity to review this research paper. This study attempted to explore Brand Personality of Korean Dance and Sustainable Behavioral Intention of Global Consumers in Four Countries usin TAM model.

The subject is appropriate and interesting. But, there are many concerns in the article that make it unsuitable for publication at its present state. The following inputs might help the author/s to improve the paper. Authors should modify the article following the comments indicated below to increase the quality of research justification, contributions, originality and findings.

First of all, paper research gap. Please improve this part in the introduction section. The introduction is very general in some sections and lacked alignment to the research findings, no discussion was provided to derive the implication from. Also, the introduction appears to be very long. Furthermore, there is insufficient support and weak arguments in support of the objective that is proposed as well as the model developed. In the final part of the introduction, the objectives proposed originality, and gaps that would be better covered. Also a summary explaining how the author will perform the methodology.

What is the originality of this research? Improve this paragraph, the paper research gap and originality should be better presented at the end of the introduction section. Please use this paper and make a citation to solve this task: https://doi.org/10.1016/j.jik.2020.08.001. Also, make notes about the manuscript structure.

There is no discussion section. This should be solved. Discussion needs to be a coherent and cohesive set of arguments that take us beyond this study in particular, and help us see the relevance of what the authors have proposed.  Author need to contextualize the findings in the literature, and need to be explicit about the added value of your study towards that literature. Also, other studies should be cited to increase the theoretical background of each of the methods used. Findings should be contextualized in the literature and should be explicit about the added value of the study towards the literature. Please use this citation to copy the style and make a citation: https://doi.org/10.1016/j.ijinfomgt.2021.102331

Questions to be answered:

What practical/professional and academic consequences will this study have for the future of scientific literature (theoretical contributions)?

Why is this study necessary? Again, the authors should make clear arguments to explain what is the originality and value of the proposed model. This should be stated in the final paragraphs of the introduction and conclusion sections.

Please consider this structure for the manuscript final part (optional)

Discussion

Conclusion

Managerial Implication

Practical/Social Implications

Limitations and future research

Please make sure your 'conclusion' section underscore the scientific value added of your paper, and/or the applicability of your findings/results, as indicated previously. Please revise your conclusion part into more details. Basically, you should enhance your contributions, limitations, underscore the scientific value added of your paper, and/or the applicability of your findings/results and future study in this section

I would also urge the authors to read the articles listed below before completing the manuscript revision. The author will understand that the article structure can be improved as well as the methodology and literature review section:

Introduction

Saura, J.R., Palacios-Marqués, D. & Iturricha-Fernández, A.  (2021). Ethical Design in Social Media: Assessing the main performance measurements of user online behavior modification. Journal of Business Research, 129, May 2021, 271-281. doi: 10.1016/j.jbusres.2021.03.001

Conclusions

Ribeiro-Navarrete, S., Saura, J. R., & Palacios-Marqués, D. (2021). Towards a new era of mass data collection: Assessing pandemic surveillance technologies to preserve user privacy. Technological Forecasting and Social Change, 167, 120681. doi: 10.1016/j.techfore.2021.120681

Good luck with your revision

Also, there is still a gap on the added value of your work in the context of proper and current research (up to 2021).

Author Response

(The authors gave the same response as above.)

Round 2

Reviewer 3 Report

Accept in present form 

Author Response

Currently, more than 232 million people around the world have been infected with “COVID-19”.

I hope you remain healthy during this prolonged COVID-19 pandemic.

Thank you very much.

Reviewer 4 Report

Most of my comments have not been applied. The responses to my comments are as well missing. I will give another chance to authors. 

Author Response

1. Revision of Introduction

We improved the "Introduction" as follows.
(extensively improved)
1. Background
2. Necessity
3. Advance Research (Limitations)
4. Originality
5. Purpose
6. Scope / Method
References
※ Other: Refer to the basic framework of the "Introduction" based on two articles

2. Revision of Discussion and Conclusion

We improved the "Discussion and Conclusion" as follows.
(extensively improved)
1. Summary
2. Implications (Business)
3. Implications (Theoretical/Social)
4. Limitations / Future Directions
References
※ Other: Refer to the basic framework of the "Discussion and Conclusion" based on two articles

Also, the “Response to Review Comments” file is uploaded on the journal website (Manuscript Submission System).
→ Please see the attachment.
